

# Effect of the molecular weight of water-soluble chitosan on its fat-/cholesterol-binding capacities and inhibitory activities to pancreatic lipase

Qiu Jin[1,2], Huahua Yu[1], Xueqin Wang[1], Kecheng Li[1] and Pengcheng Li[1]

[1] Key Laboratory of Experimental Marine Biology, Institute of Oceanology, Chinese Academy of Sciences, Qingdao, Shandong, China
[2] University of Chinese Academy of Sciences, Beijing, China

## ABSTRACT

**Background**. Obesity has become a worldwide burden to public health in recent decades. Given that obesity is caused by an imbalance between caloric intake and expenditure, and that dietary fat is the most important energy source of all macronutrients (by providing the most calories), a valuable strategy for obesity treatment and prevention is to block fat absorption via the gastrointestinal pathway. In this study, the fat- and cholesterol-binding capacities and the inhibition of pancreatic lipase by water-soluble chitosan (WSC) with different weight-average molecular weight (Mw) were tested and compared *in vitro*, in order to determine the anti-obesity effects of WSC and the influence of its Mw.

**Methods**. In this study, WSC with different Mw ($\sim$1,000, $\sim$3,000, $\sim$5,000, $\sim$7,000 and $\sim$9,000 Da) were prepared by oxidative degradation assisted with microwave irradiation. A biopharmaceutical model of the digestive tract was used to determine the fat- and cholesterol-binding capacity of WSC samples. The pancreatic lipase assays were based on p-nitrophenyl derivatives.

**Results**. The results showed that all of the WSC samples exhibit great fat- and cholesterol-binding capacities. Within the testing range, 1 g of WSC sample could absorb 2–8 g of peanut oil or 50–65 mg of cholesterol, which are both significantly higher than the ability of cellulose to do the same. Meanwhile, all the WSC samples were proven to be able to inhibit pancreatic lipase activity to some extent.

**Discussion**. Based on the results, we suggest that there is a significant correlation between the binding capacity of WSC and its Mw, as WSC2 ($\sim$3,000 Da) shows the highest fat- and cholesterol-binding capacities (7.08 g $g^{-1}$ and 63.48 mg $g^{-1}$, respectively), and the binding ability of WSC declines as its Mw increases or decreases from 3,000 Da. We also suggest WSC as an excellent resource in the development of functional foods against obesity for its adsorption, electrostatic binding and entrapment of cholesterol, fat, sterols and triglycerides in the diet.

Corresponding authors
Huahua Yu, yuhuahua@qdio.ac.cn
Pengcheng Li, pcli@qdio.ac.cn

## INTRODUCTION

The prevalence of obesity, caused by an imbalance between caloric intake and expenditure, has dramatically risen in recent years (*Racette, Deusinger & Deusinger, 2003*; *Ogden et al., 2014*). According to the WHO report of 2009, more than one billion people are overweight worldwide, and at least 400 million of them are defined as obese (*Rodgers, Tschöp & Wilding, 2012*). Obesity was defined as a worldwide chronic disease by WHO in 2004 and has been proven to be a major risk factor for developing other diseases such as hyperlipidemia, hypertension, type 2 diabetes, cardiovascular diseases, and certain cancers (*Saltiel & Kahn, 2001*; *Rahmouni et al., 2005*; *Vucenik & Stains, 2012*; *Bastien et al., 2014*; *Tobias et al., 2014*). One of the principal research interests in food science in recent years has been the extraction and identification of natural-derived bioactive compounds with anti-obesity effects.

Chitosan (CTS), a polyglucosamine derived from chitin, is one of these attractive compounds. As the second most abundant polysaccharide in nature, chitosan locates mainly in the exoskeletons of arthropods, such as crabs, lobsters, shrimps and insects (*Hu et al., 2016*). As a dietary fiber, chitosan cannot be digested by the digestive enzymes of humans. Although the specific mechanism remains controversial, there are already lots of studies indicate that chitosan is effective in obesity treatment. For example, a mixture of chitin (20%) and chitosan (80%) was proved to be able to prevent the increase of body weight of diet-induced obese mice via enhancing fat excretion and inhibiting lipid absorption (*Han, Kimura & Okuda, 1999*). In a study of human obesity, the chitosan group demonstrated more body weight loss and improved body composition index, compared with the placebo control (*Kaats, Michalek & Preuss, 2006*). However, there are also studies indicating that taking water-insoluble chitosan with high molecular weight (Mw) simply would cause side effects such as nausea and constipation (*Sumiyoshi & Kimura, 2006*; *Neyrinck et al., 2009*; *Qinna et al., 2013*). Water-soluble chitosan (WSC) is a derivative of chitosan. Studies have proven that WSC has a higher reactivity and fewer side effects than water-insoluble chitosan in reducing food absorption (*Choi et al., 2002*). Meanwhile, the study of *Choi et al. (2002)* also reported an improvement in the ovarian and oviduct dysfunction in mice that were fed a high-fat diet. Due to its significant advantages, including biocompatibility, nontoxic nature and high solubility in neutral aqueous solutions, WSC is considered as an ideal lipid-lowering dietary supplement.

The weight-average molecular weight (Mw) of chitosan is an important characteristic that greatly affects its chemical and physiological properties. The Mw of chitosan is proportional to its viscosity. And there are studies proving that viscous polysaccharides can cause entrapments, which would reduce the absorption of fat and cholesterol in the diet (*Kanauchi et al., 1995*). However, given the wide range of chitosan Mw, studies about the relationship between the Mw of chitosan and its anti-obesity effect are less reported and unsystematic. There have been studies suggesting that low-Mw chitosan oligosaccharide (COS) (1,000–3,000 Da) is more effective than high-Mw COS in anti-obesity function by inhibiting adipocyte differentiation in 3T3-L1 cells (*Kumar et al., 2009*; *Rahman et al., 2008*). In addition, the results of another study suggested that the 46,000 Da chitosan was more effective than chitosan of 21,000 or 13,0000 Da in anti-obesity function by inhibiting

pancreatic lipase activity (*in vitro*) and plasma triacylglycerol elevation in the oral lipid tolerance test (*Sumiyoshi & Kimura, 2006*). Therefore, further research is necessary to systematically investigate the relationship between chitosan Mw and its anti-obesity effect.

Current therapies of obesity mainly include appetite control, blocking fat absorption, stimulating energy expenditure, suppressing adipose tissue growth and increasing body fat mobilization (*Hu et al., 2016*). Dietary fat is the most important energy source of all macronutrients, providing the most calories. It has been demonstrated that there is a direct relation between dietary fat intake and obesity onset (*Hu et al., 2016*). Therefore, it is a valuable strategy for obesity treatment and prevention by blocking fat absorption via the gastrointestinal pathway. It has been proven that sufficient fiber, such as cellulose, in the diet could reduce weight by preventing excessive fat intake and fat accumulation in adipose depots (*Van Itallie, 1978*). WSC has also been suggested as an ideal lipid-lowering dietary supplement due to its effective fat-/cholesterol-binding capacity, biocompatibility, nontoxic nature and facile solubility in neutral aqueous solutions (*Anraku et al., 2009*). Pancreatic lipase (PL) is a key enzyme for the absorption of dietary triglycerides. It rapidly converts a triglyceride into a 2-monoglycerol and two free fatty acids (*Gu et al., 2011*). Given the key role PL plays in starch and lipid digestion, it represents an attractive target for the prevention of excessive body weight gain. Orlistat, a potent competitive inhibitor of PL, is currently available as an anti-obesity drug. Although it has been reported that orlistat showed a significant weight-reducing effect over both short- and long-term periods, there were also some harmful gastrointestinal side effects reported, including loose stools and oily stools/spotting (*Derosa et al., 2005*). WSC was also proven to be able to inhibit pancreatic lipase, and the inhibitory effect was suggested to be related to its Mw (*Tsujita et al., 2007*). However, the underlying mechanism is still not understood, and there are few related reports; thus, further studies are needed.

In the present study, the fat-binding and cholesterol-binding capacities of WSC with different Mw, as well as its inhibitory effects on PL, were tested and compared in order to determine the anti-obesity effects of WSC and the influence of its Mw.

# MATERIALS AND METHODS

## Materials and equipment

Chitosan (referred as CTS0, Mw of ~1,800,000 Da, the degree of deacetylation was 82%) from shrimp shell was purchased from Baicheng Biochemical Corp. (Qingdao, China). The microwave synthesis/extraction reaction station was purchased from SINEO Microwave Chemistry Technology Co., Ltd. (Shanghai, China). The oscillation incubator (ZQLY-180F) was purchased from Shanghai Zhichu Instruments Co., Ltd. (Shanghai, China). The total cholesterol assay kit was purchased from Nanjing Jiancheng Bioengineering Institute. Lipase (from beef pancreas, powder, 15–35 units/mg) was purchased from the Aladdin Industrial Corporation (Shanghai, China). $H_2O_2$ and other reagents were of analytical reagent grade.

## Preparation of WSC with Mw of ~1,000, ~3,000, ~5,000, ~7,000 and ~9,000 Da

WSC samples with Mw of ~1,000, ~3,000, ~5,000, ~7,000 and ~9,000 Da were prepared by the method of *Li et al. (2012)* with some modifications. In brief, 5 g of CTS0 was introduced to 250 mL of 2% acetic acid, and then 5 mL of 30% $H_2O_2$ was added. Afterwards, the microwave-assisted degradation was carried out at a power of 600 W at 70 °C for 60, 32, 30, 22 and 18 min, respectively. Immediately after the reaction, the reaction mixture was cooled to room temperature, and the pH was adjusted to 7.0 with aqueous NaOH solution (10 mol/L). The degraded products were then precipitated by adding ethanol. Finally, the precipitate was collected by centrifugation for 10 min at 3,740 × g and lyophilized to yield powdered products. The products that were degraded for 60, 32, 30, 22 and 18 min were referred as WSC1–5, respectively.

## Characterization

The weight-average molecular weight (Mw) of WSC1-5 were measured by the method of *Zou et al. (2015)* with some modifications. In brief, the Mw was measured by an Agilent 1,260 gel permeation chromatography (Agilent Technologies, Santa Clara, CA, USA) equipped with a refractive index detector. Chromatography was performed on TSK G3000-PWXL columns at a column temperature of 30 °C, a flow rate of 0.8 mL/min, and a $CH_3COOH$ (0.2 mol/L)/ $CH_3COONa$ (0.1 mol/L) aqueous solution was used as mobile phase. The sample concentration was 0.3% (w/v). Dextrans (Sigma, St. Louis, MO, USA) with Mw of 1,000, 5,000, 12,000, 25,000 and 50,000 Da were used to calibrate the column.

Fourier transform infrared (FT-IR) spectra of CTS0 and the degraded chitosan were measured in the 4,000–400 cm$^{-1}$ regions using a Thermo Scientific Nicolet iS10 FT-IR spectrometer in KBr discs.

## Estimation of the fat-binding capacity of WSC

The fat-binding capacity was evaluated *in vitro* by the method of *Zhang et al. (2012)* with some modifications. Briefly, 120 mg of WSC was mixed with 10 mL of HCl (0.1 mol/L). After addition of 10 g of peanut oil, the mixture was swirled thoroughly and incubated at 37 °C with shaking (300 rpm). After 2 h of incubation, the pH of the mixture was adjusted to the range of 7.0–7.60 by adding NaOH (1 mol/L). Then, the resulting mixture was incubated for another 2 h at 37 °C and 300 rpm. Afterwards, it was cooled to room temperature and centrifuged at 3,740 × g for 20 min. The supernatant oil, considered an unbound oil, was carefully and quantitatively removed. To release the oil bound by WSC while it was in alkaline solution and to dissolve the WSC, HCl solution (0.1 mol/L) was added to the WSC-water layer until the pH was 3.0. Afterwards, double extraction with ethyl ether was used to remove the released oil. Then, the combined ether extracts were kept at 40 °C until the ether had completely evaporated. At last, the remaining oil was weighed, and this mass was used to calculate the fat-binding capacity of WSC. The fat-binding capacity of the WSC sample was expressed as grams of bound oil per gram of WSC. Cellulose was used as a positive control, and the solution without WSC was used as a substrate blank. The test was conducted in triplicate. Scanning electron microscope

(SEM) image and FT-IR spectra were used to identify the structure and composition of the precipitate that was collected after centrifugation.

## Estimation of the cholesterol-binding capacity of WSC

The cholesterol-binding capacity was evaluated *in vitro* by the method of *Liu, Zhang & Xia (2008)* with some modifications. First, a cholesterol micellar solution was prepared by sonication; every 1 mL sample contained 10 mM sodium taurocholate, 2 mM cholesterol, 5 mM oleic acid, 132 mM NaCl, and 15 mM sodium phosphate buffer (pH 7.4). Then, 60 mg of WSC was added to 5 mL of the micellar solution. The mixture was incubated for 2 h in a 37 °C shaker bath. Then, the mixture was transferred to a centrifuge tube and centrifuged at 23,294 × g for 20 min at 37 °C. The supernatant was collected for the determination of cholesterol. The amount of binding was calculated as the amount of cholesterol in the supernatant of the substrate blank subtracted from the amount in the supernatant of the sample. The amount of cholesterol was determined by the total cholesterol assay kit. The binding capacity of WSC was calculated as the milligrams of bound cholesterol per gram of WSC. Cellulose was used as a positive control, while the micellar solution without WSC was used as a substrate blank. The test was conducted in triplicate. SEM image and FT-IR spectra were used to identify the structure and composition of the precipitate that was collected after centrifugation.

## Assay of the inhibitory effects of WSC on the pancreatic lipase

The assay of the inhibitory effect on pancreatic lipase *in vitro* was measured by the method of *Margesin et al. (2002)* with some modifications. First, a calibration curve of the concentration and absorbance at 405 nm of pNP was prepared. Standards containing 0 (reagent blank), 25, 50, 75, 100 and 125 µg of pNP were made by adjusting 0 to 1.25 mL of a working standard solution to 5 mL with buffer. The absorption of pNP was measured spectrophotometrically at 405 nm against the reagent blank. A calibration curve relating the concentration and absorbance of pNP was obtained. Then, 700 µL of Tris-HCL buffer (50 mM, pH 8), 100 µL of WSC sample and 100 µL of pancreatic lipase solution of a certain concentration were prewarmed at 37 °C in a water bath for 10 min. Afterwards, 100 µL of substrate solution (a certain amount of p-nitrophenyl butyrate (pNPB) that was diluted in dimethyl sulfoxide (DMSO) and stored at −20 °C) were added. The contents were mixed, and the tubes were incubated in the water bath at 37 °C for exactly 15 min. Then, the tubes were cooled for 10 min on ice immediately to stop the reaction. Afterwards, the tube contents were centrifuged at 3,740 × g and in the range of 2–4 °C for 5 min. The supernatants were pipetted into test tubes that were held on ice. Immediately afterwards, the absorption of the released pNP was measured spectrophotometrically at 405 nm against the reagent blank. The concentration of pNP could be obtained by referencing the calibration curve. Orlistat was used as a positive control. One unit of PL activity was defined as the amount of the enzyme that releases 1 µmol of pNP in 1 min under certain conditions (U/mL). The activity of PL was calculated by the following formula:

$$\text{PL activity} = \frac{CV}{TV'}$$

where PL activity is one unit of PL activity, in U/mL; $C$ is the concentration of pNP, in $\mu$mol/mL; $V$ is the final volume of the reaction, in mL; $T$ is the reaction time, in min; and $V'$ is the volume of the PL, in mL.

The inhibition rate (%) was calculated by the following formula:

$$\text{Inhibition rate (\%)} = 100 - \left( \frac{B - b}{A - a} \times 100 \right)$$

where $A$ is the PL activity without inhibitor, $a$ is the negative control without inhibitor, $B$ is the PL activity with inhibitor, and $b$ is the negative control with inhibitor.

The reaction speed (mmol/L/s) was calculated by the following formula:

$$v = \frac{A_{405 \text{ nm}}}{\varepsilon \times 10^{-3} \times t}$$

where $v$ is the reaction speed, in mmoL/L/s; $A_{405 \text{ nm}}$ is the absorbance of the released pNP measured spectrophotometrically at 405 nm; $\varepsilon$ is the extinction coefficient of the released pNP, in L/(mol cm). The extinction coefficient of pNP at 405 nm is 18,800 (L/moL/cm); $t$ is the reaction time, in s.

### Statistical analysis

All data are expressed as the mean $\pm$ SD. Differences between groups were determined by one-way analysis of variance, using SPSS 17.0. Results were considered significant if the value of $p$ was <0.05.

## RESULTS AND DISCUSSION

### Degradation of chitosan by hydrogen peroxide under microwave irradiation

The degradation products of chitosan are complex mixtures. The weight-average molecular weights (Mw) of WSC1-5 were measured by high-performance liquid chromatography (HPLC). HPLC determination of Mw could standardize the mixtures and provide relative quantitative parameters that could feasibly be used for comparisons. Figure 1 shows changes of Mw of chitosan with different reaction times. These tests were carried out in the presence of $H_2O_2$ at 70 °C and a microwave power of 600 W. CTS0, with Mw of ~1,800,000 Da, was used as an ingredient. The Mw of degraded products decreased quickly at the beginning and could reach ~9,000 Da in 18 min. WSC with a low Mw of ~1,000 Da were obtained when the degradation time reached 60 min. In addition, at that point, the change in Mw was no longer obvious. The HPLC figures show that the Mw of the degradation products after 60, 32, 30, 22 and 18 min (referred as WSC1–5) of irradiation were 1294.3, 3135.6, 4951.9, 7312.8 and 9304.3 Da, respectively (Fig. 2).

Figure 3 showed the FT-IR spectra of the degraded chitosan and the original chitosan (CTS0). It could be observed that the characteristic absorption bands of CTS0 were appeared at 1641.29 cm$^{-1}$ (Amide I), 1599.81 cm$^{-1}$ ($NH_2$ bending) and 1327.60 cm$^{-1}$ (Amide III). The bands in the range 1,158–895 cm$^{-1}$ were assigned to the characteristics of its polysaccharide structure (*Peniche et al., 1999*). Compared with the FT-IR spectra of CTS0, the spectra of the degraded chitosan exhibited most of the bands as CTS0, indicating

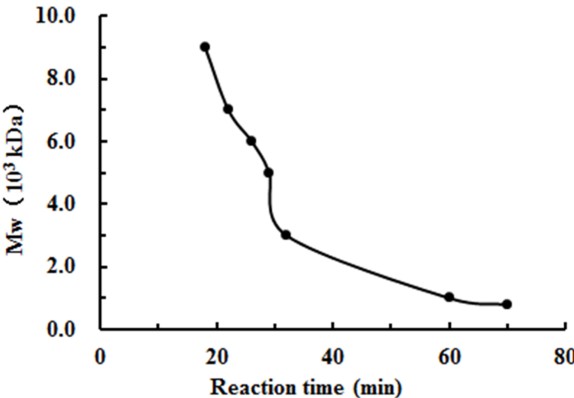

**Figure 1 The effect of reaction time on the degradation of chitosan under the conditions of $H_2O_2$ assisted with microwave irradiation.** CTS0 (5 g) was introduced to 250 mL of 2% acetic acid, and then 5 mL of 30% $H_2O_2$ was added. Afterwards, assisted with microwave radiation, the degradation was carried out at a power of 600 W at 70 °C for 60, 32, 30, 22 and 18 min, respectively. The weight-average molecular weight was measured by HPLC.

that the main polysaccharide structure of the degraded chitosan still remained. However, there still existed some differences between the degraded chitosan and CTS0. For example, bending and stretching of N-H and O-H shifted to low wave number, which indicated that the intermolecular and intramolecular hydrogen bonds of chitosan were weakened. Meanwhile, 1327.60 cm$^{-1}$ (Amide III) also shifted to low wave number, indicating that with the decrease of the molecular weight of chitosan, the free amino group numbers of degraded products decreased.

WCS with different Mw can be obtained by a variety of techniques, such as acid hydrolysis, oxidative degradation and enzymatic methods (*Chang, Tai & Cheng, 2001*; *Muzzarelli et al., 2002*; *Trombotto et al., 2008*; *Aam et al., 2010*). All of these techniques have advantages as well as disadvantages. For example, the acid hydrolysis of chitosan not only requires special equipment with corrosion resistance to concentrated acid but also has major waste disposal problems. The enzymatic methods do not require special equipment, but the cost of enzymes is rather high. Oxidative degradation is easily available and environmentally friendly. The depolymerization caused by free radical reactions dominates. However, the use of $H_2O_2$ alone makes the formation of free radicals inefficient, resulting in a slow degradation of chitosan. Meanwhile, there are some side reactions that can change the chemical structure of chitosan. In recent years, microwave-assisted oxidative degradation has received increasing attention due to its remarkable advantages, including lower concentrations of $H_2O_2$, lower temperatures and shorter reaction times compared with conventional heating modes (*Li et al., 2012*).

## Fat- and cholesterol-binding capacity of WSC

In this study, five WSC samples with the same degree of deacetylation and different Mw (~1,000, ~3,000, ~5,000, ~7,000, and ~9,000 Da) were used to identify the relationship between the fat-/cholesterol-binding capacity of WSC and its Mw (<10,000 Da). A biopharmaceutical model of the digestive tract was used to show the fat- and cholesterol-binding

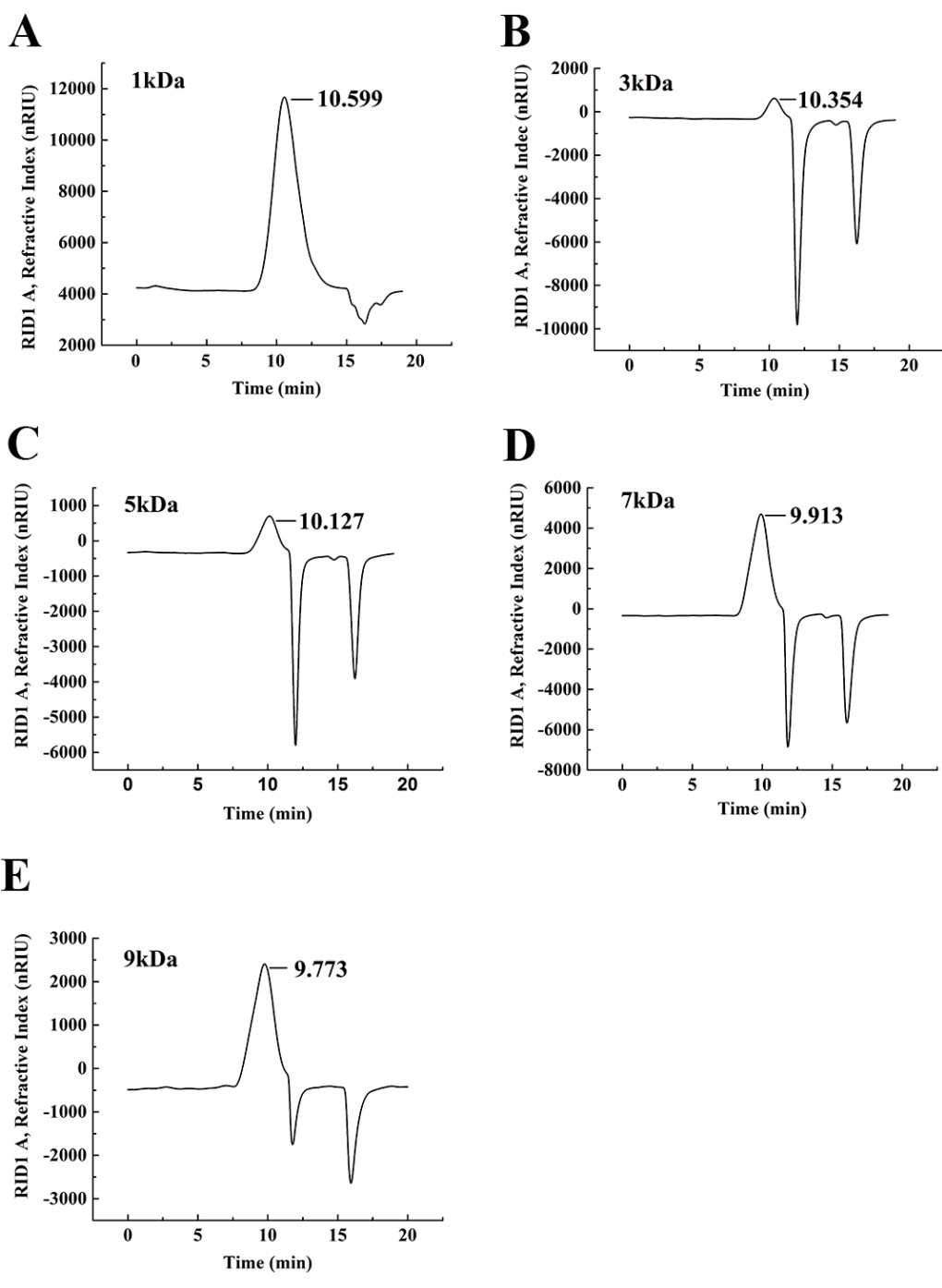

**Figure 2  The weight-average molecular weight assay of WSC1-5 with HPLC.** The retention time and weight-average molecular weight of (A) WSC1, (B) WSC2, (C) WSC3, (D) WSC4 and (E) WSC5 were measured by an Agilent 1,260 gel permeation chromatography equipped with a refractive index detector. Chromatography was performed on TSK G3000-PWXL columns, using 0.2 M $CH_3COOH/0.1MCH_3COONa$ aqueous solution as the mobile phase at a flow rate of 0.8 mL/min with a column temperature of 30 °C. The sample concentration was 0.3% (w/v). The standards used to calibrate the column were dextrans with Mw of 1,000, 5,000, 12,000, 25,000 and 50,000 Da.

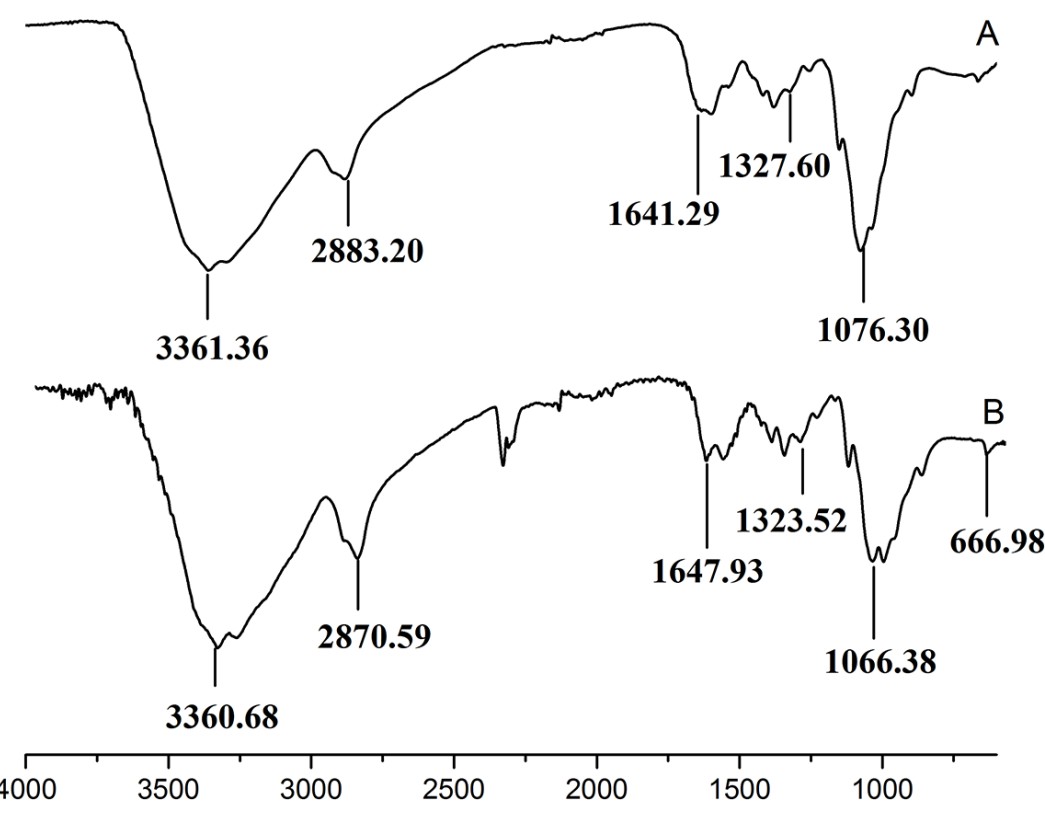

**Figure 3** FT-IR spectra of (A) original chitosan, (B) the degraded chitosan.

capacity of WSC samples. As the results show, within the testing range, 1 g of WSC could absorb 2–8 g of peanut oil (Fig. 4) or 50–65 mg of cholesterol (Fig. 5), of which both numbers are significantly higher than that of cellulose. WSC2 showed the highest peanut oil-binding capacity, which was 7.08 g g$^{-1}$, which is much higher than its affinity for cellulose (0.41 g g$^{-1}$). The peanut oil-binding capacity of WSC3 was the second highest and was not significantly different from that of CTS0. In addition, there were no pronounced differences between the capacities of WSC1, WSC4 and WSC5. For the cholesterol-binding capacity, a similar trend was that WSC2 and WSC3 showed the highest cholesterol-binding capacities of 63.48 mg g$^{-1}$ and 62.91 mg g$^{-1}$, respectively. WSC1 followed, while WSC4 and WSC5 had the least capacity. It was worth mentioning that all the WSC samples showed higher cholesterol-binding capacities than CTS0, and all the chitosan samples, including CTS0 and WSC1-5, showed significantly higher cholesterol-binding capacities than cellulose. The SEM image shows that WSC2 has a homogeneous and irregular shape (Fig. 6A), while the SEM images of the precipitate formed by WSC2 and peanut oil (Fig. 6B) or cholesterol (Fig. 6D) present irregular, rugged, nubbly shapes. Obvious microspheres were also seen (Figs. 6C and 6E). The FT-IR spectra of WSC2, peanut oil and their precipitate are shown in Fig. 7, while the FT-IR spectra of WSC2, cholesterol and their precipitate are shown in Fig. 8. The characteristic absorption bands of WSC2 appeared at 1574.32 cm$^{-1}$ (NH$_2$ bending vibration) and 1415.31 cm$^{-1}$ (Amide III). The bands in the range of 1158–895 cm$^{-1}$ were assigned to characteristics of the polysaccharide structure (Figs. 7A and 8A) (*Peniche et al.,*

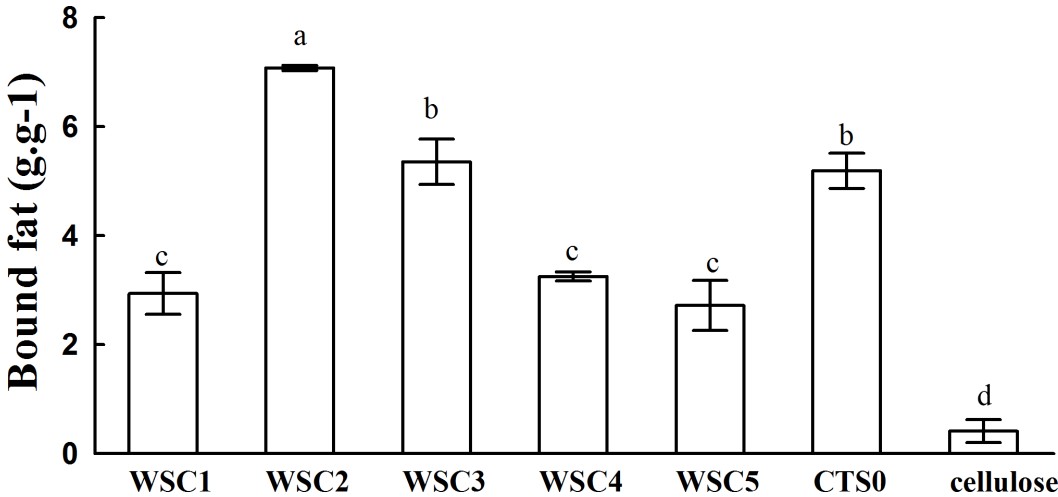

**Figure 4** **Fat-binding capacities of WSC1-5, CTS0 and cellulose _in vitro_.** The mixture of WSC (or chitosan), HCl and peanut oil was swirled thoroughly and incubated for 2 h at 37 °C and 300 rpm. Then, the pH was adjusted to the range of 7.0–7.60 and incubated for another 2 h at 37 °C and 300 rpm. Afterwards, the sample was cooled to room temperature and centrifuged at 3,740 × g for 20 min. Then, the supernatant oil was removed, and HCl solution was added until the pH was 3.0. After double extraction of the sample with ethyl ether, the combined ether extracts were kept at 40 °C until complete evaporation of ether. The remaining oil was weighed. The fat-binding capacity of the WSC sample was expressed as grams of bound oil per gram of WSC. Cellulose was used as a positive control, and the solution without WSC was used as a substrate blank. Data are expressed as the mean ± SD from triplicate experiments. Different small letters next to values indicate significant differences ($p < 0.05$).

_1999_). The characteristic absorption bands of peanut oil appeared at 2,970–2,850 cm$^{-1}$ (C-H symmetric or asymmetric.), 1746.49 cm$^{-1}$ (RC = OOR stretching vibration) and 1162.90 cm$^{-1}$ (C-C stretching vibration) (Fig. 7B) (_Alexa et al., 2009_). The characteristic absorption bands of both WSC2 and peanut oil were observed in the precipitate (Fig. 7C), proving that both WSC2 and peanut oil are included in the precipitate. The characteristic absorption bands of cholesterol appeared at 3297.10 cm$^{-1}$ (O-H stretching vibration) and 1052.37 cm$^{-1}$(C-O stretching vibration). The bands in the ranges of 3,000–2,850 cm$^{-1}$ and 1,464–1,340 cm$^{-1}$ were assigned to the characteristics of the C-H (Fig. 8B) (_Deleris & Petibois, 2002_). The FT-IR spectra of the precipitate (Fig. 8C) shows that there are characteristic absorption bands of both WSC2 and cholesterol, indicating that both WSC2 and cholesterol are included in the precipitate. According to all these results, we suggest that WSC has the ability to form micelles and to trap the oil phase and cholesterol upon precipitation.

Electrostatic effects, embedding and adsorption are usually considered the three main mechanisms of chitosan's fat- and cholesterol-binding capacities (_Czechowska-Biskup et al., 2005_; _Liu, Zhang & Xia, 2008_). As a weak cationic polyelectrolyte, chitosan is soluble in aqueous solutions of inorganic and organic acids, forming strongly charged macromolecules. Therefore, when it is eaten together with fat and cholesterol, in the condition of the stomach with a soluble pH 2, the ionized polycationic chitosan would form complexes and micelles with negatively charged molecules and particles, such as triglycerides, fatty and bile acids, cholesterol and other sterols. As a result, the complexes and micelles would not be absorbed. The molecular conformation, charge density and

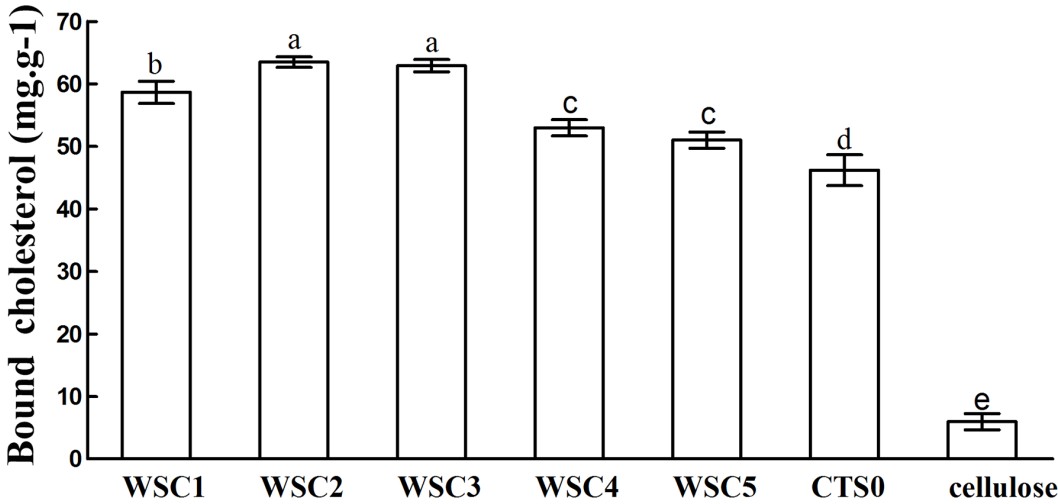

**Figure 5   Cholesterol-binding capacities of WSC1-5, CTS0 and cellulose *in vitro*.** The mixture of WSC (or chitosan) and the cholesterol micellar solution was incubated for 2 h in a 37 °C shaker bath and then centrifuged at 23,294 ×g for 20 min at 37 °C. The amount of binding was calculated as the amount of cholesterol in the supernatant of the substrate blank subtracted from the amount in the supernatant of the sample. The amount of cholesterol was determined by the total cholesterol assay kit. The binding capacity of WSC was calculated as milligrams of bound cholesterol per gram of WSC. Cellulose was used as a positive control, while the micellar solution without WSC was used as a substrate blank. Data are expressed as the mean ± SD from triplicate experiments. Different small letters next to values indicate significant differences ($p < 0.05$).

distribution along the chain will contribute to the effectiveness of this binding. However, due to the deprotonation of the amino groups, chitosan precipitates when the pH rises above 6.5. So, as digestion progresses and the pH rises in the solutions of duodenum and intestine, chitosan molecules lose their charge and precipitate with trapped micelles and fat microdroplets. As a result, the excretion of fatty materials, including cholesterol, sterols and triglycerides, is promoted. This conclusion is verified by the studies of *Xia et al. (2011)*. By measuring the content of fluorescein-isothiocyanate-labeled chitosan (FITC-CIS) in plasma and tissues following oral administration of FITC-CIS in mice, *Xia et al. (2011)* deduced that chitosan could directly bind dietary fat in the digestive tract and then be excreted with the feces, and a fraction of the chitosan was degraded into chitosan oligosaccharide to regulate lipid metabolism. Sufficient fiber, such as cellulose, in the diet was accepted by the public as an ideal diet food that reduces weight (*Van Itallie, 1978*; *Howarth, Saltzman & Roberts, 2001*; *Slavin, 2005*; *Anraku et al., 2009*). The anti-obesity effect of cellulose mainly depends on decreasing the caloric density of the diet, slowing the rate of food ingestion, increasing the effort involved in eating, promoting intestinal satiety and interfering slightly with efficiency of energy absorption. Its adsorption ability *in vitro* mainly depends on having physical adsorption that is much weaker than that of chitosan. In contrast, chitosan shows a higher fat- and cholesterol-binding capacity than cellulose *in vitro*.

The result also indicates that the Mw of WSC plays an important role in its fat- and cholesterol-binding capacity. Although no convincing data have been published that present a straightforward dependence between the fat-/cholesterol-binding ability of chitosan and

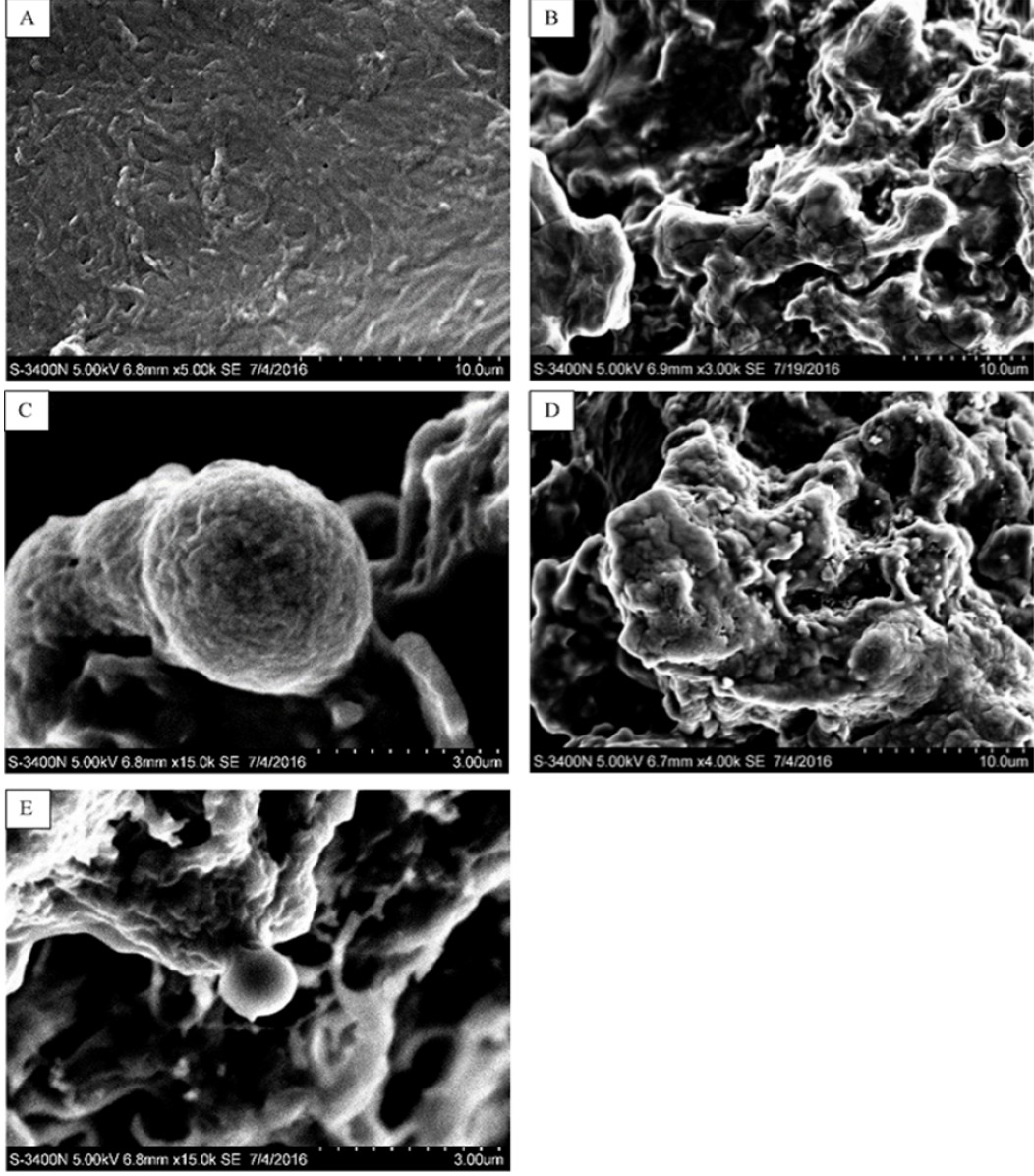

**Figure 6  SEM images of WSC2, the precipitate formed by WSC2 and peanut oil, and the precipitate formed by WSC2 and cholesterol.** SEM of (A) WSC2, (B) the precipitate formed by WSC2 and peanut oil, (C) obvious microspheres in the precipitate formed by WSC2 and peanut oil, (D) the precipitate formed by WSC2 and cholesterol, and (E) obvious microspheres in the precipitate formed by WSC2 and cholesterol.

its Mw, previous studies have suggested a general tendency that lowering the Mw of chitosan leads to an increase in their fat-/cholesterol-binding capacity (*Wojtasz-Pajak et al., 1998*; *Czechowska-Biskup et al., 2005*). On the one hand, shorter chains, which have a higher mobility (such as WSC2 (∼3,000 Da)), may be more effective in forming or occluding micelles. In addition, when the chain becomes longer, such as the cases of WSC4–5 (∼7,000–9,000 Da), the mobility decreases, which is disadvantageous for micelle

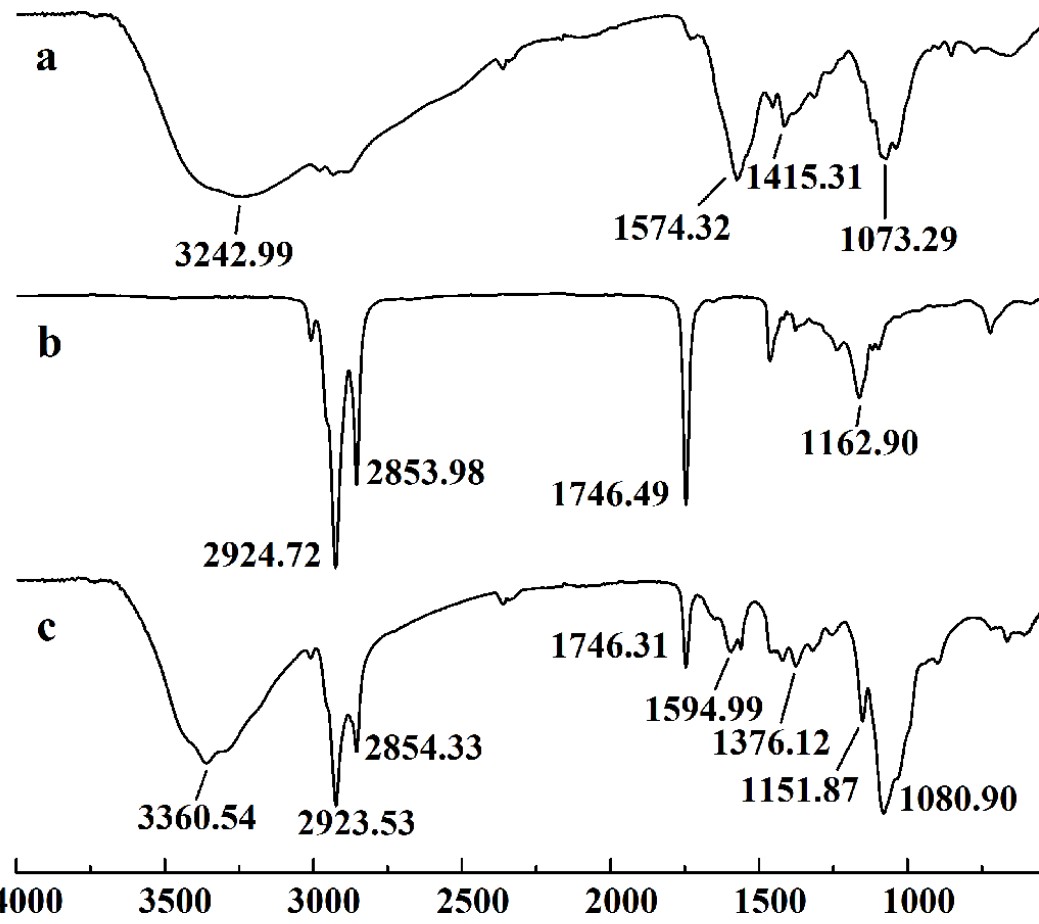

**Figure 7** **FT-IR spectra of WSC2, peanut oil and the precipitate collected after centrifugation.** The FT-IR spectra of (A) WSC2, (B) peanut oil, and (C) the precipitate collected after centrifugation.

formation. However, on the other hand, if the chains become too short, such as the case of WSC1 (~1,000 Da), their solubility at pH 7 would increase. As a result, they would probably tend to form individual solubilized molecules rather than oil- and cholesterol-trapping micelles and/or matrices. This hypothesis was verified by this study, as WSC2 (~3,000 Da) shows the highest binding capacity. In addition, the binding capacity decreased as the Mw increased or decreased.

## Inhibition of pancreatic lipase
### Estimation of the optimum reaction temperature, pH, substrate concentration, enzyme concentration and the addition order of the substrate, enzyme, and inhibitor

The appropriate methodological strategy to characterize an inhibitor depends on the type of inhibitor. In addition, it is particularly important to know which substrate concentrations constitute saturating conditions for the enzyme. Therefore, in order to identify the PL inhibitory ability of WSC accurately, the optimum reaction temperature, pH, substrate concentration, enzyme concentration and the addition order of substrate, enzyme and inhibitor were determined (Fig. 9).

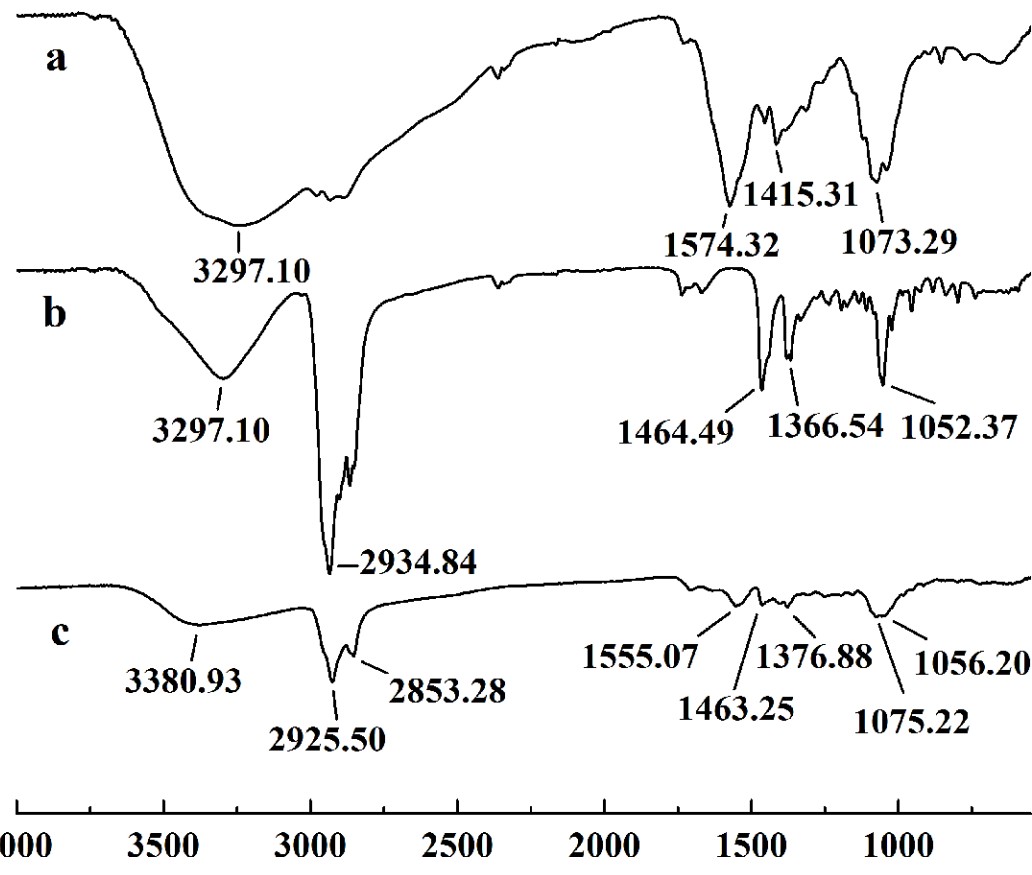

**Figure 8** **FT-IR spectra of WSC2, cholesterol and the precipitate collected after centrifugation.** The FT-IR spectra of (A) WSC2, (B) cholesterol, and (C) the precipitate collected after centrifugation.

As Fig. 9A showed, an increase in the temperature of incubation from 30 to 37 °C significantly increased lipase activity (U/mL) The highest activity was measured at 37 °C. To standardize the method, a temperature of 37 °C was chosen because the most precise result (lowest standard deviations) was obtained at this temperature; 37 °C was also the normal body temperature at which pancreatic lipase played a role.

Lipase activity (U/mL) increased with increasing pH in the range from 6 to 8. At pH 9.0, a rapid decrease of enzyme activity due to enzyme denaturation was observed. Enzyme assays based on p-nitrophenyl derivatives were usually carried out at a pH of 7.25 to 8.0 according to other studies (*Feller et al., 1991*; *Ishimoto, Sugimoto & Kawai, 2001*). In this study, pH 8.0 was used as the optimum pH, as Fig. 9B showed.

The relationship between substrate concentration and reaction speed was measured in this study. An increase in the substrate concentration from 2–14 $\times 10^{-4}$ mol/L increased reaction speeds, as Fig. 9C showed. However, when the substrate concentration was more than $10 \times 10^{-4}$ mol/L, the change was no longer apparent. To standardize the method, a substrate concentration of $10 \times 10^{-4}$ mol/L was chosen for all further analyses.

As Fig. 9D showed, there was an increase of the reaction speed when the enzyme concentration increased from 0.1 to 0.7 mg/mL. In addition, when the lipase concentration

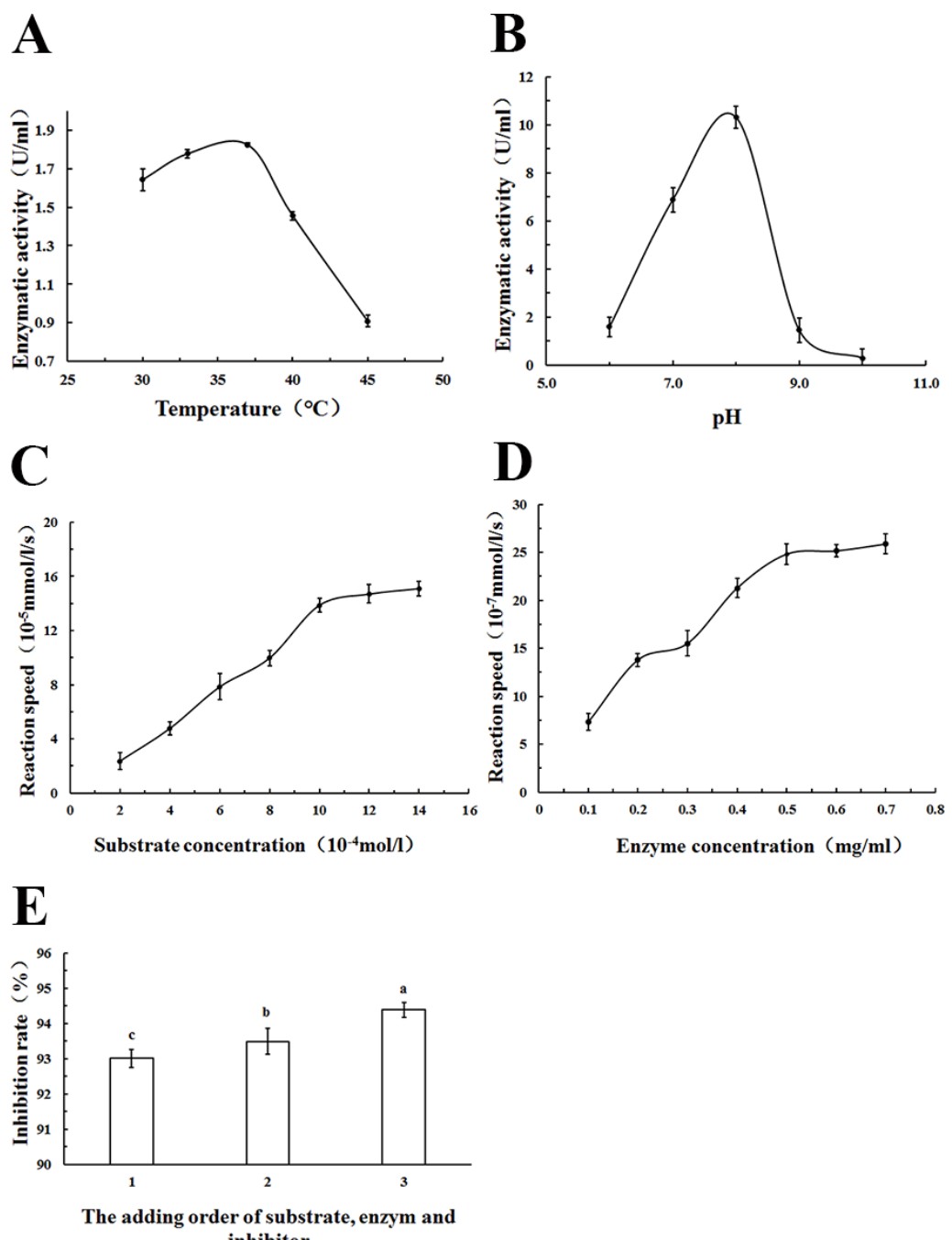

**Figure 9  Estimation of the optimum reaction conditions for PL.** The effect of (A) reaction temperature, (B) reaction pH on PL enzymatic activity (U/mL), (C) substrate concentration, and (D) enzyme concentration on reaction speed ($10^{-5}$ mmol/l/g), and the addition order of reagents on PL activity inhibition rate (%) (1: substrate and enzyme were prewarmed at 37 °C for 10 min first, and then the inhibitor was added; 2: substrate and inhibitor were prewarmed at 37 °C for 10 min first, and then the substrate was added) were determined by colorimetric methods using pNPB) as a chromogenic substrate. Data are expressed as the mean ± SD from triplicate experiments. Different small letters next to values indicate significant differences ($p < 0.05$).

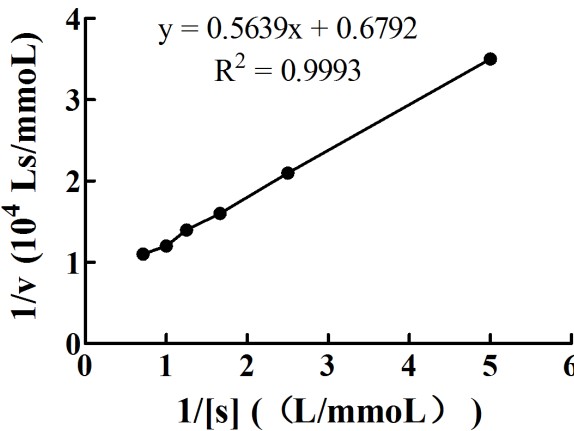

**Figure 10 Double reciprocal plot of 1/[S] (L/mmol) and 1/v ($10^4$ Ls/mmol).** PL and pNPB were mixed and incubated in the water bath at 37 °C for exactly 15 min. Then, immediately, the contents were cooled for 10 min on ice to stop the reaction and centrifuged at 3,740 × g and in the range of 2–4 °C for 5 min. The supernatants were pipetted to measure the absorbance of the released pNP spectrophotometrically at 405 nm against the reagent blank. Data are expressed as the means ± SD from triplicate experiments. Different small letters next to values indicate significant differences ($p < 0.05$).

was more than 0.5 mg/mL, the change became no longer apparent. Therefore, a lipase concentration of 0.5 mg/mL was chosen as the optimum concentration for all further analyses.

Using different addition orders of the substrate, enzyme and inhibitor (orlistat) showed different inhibition rates (%) of pancreatic lipase (Fig. 9E). The biggest inhibition rate was shown when orlistat and lipase were prewarmed at 37 °C for 10 min before the addition of the substrate. This addition order ensured that there was adequate contact between the inhibitor and the lipase. This order was used in all further analyses.

$K_m$, a kinetic constant, is determined by the enzyme and is not affected by the concentration. It can be obtained from Michaelis–Menten equation. Figure 10 shows a double reciprocal plot of 1/[S] (L/mmol) and 1/v ($10^4$ Ls/mmol). The absolute value of the $x$-intercept is $1/K_m$. The $K_m$ of PL calculated by our study is $1.20 \times 10^{-3}$ mol/L. The equation of the line fitting the data in this plot is $y = 0.5639x + 0.6792$, with $R^2 = 0.9993$.

### Inhibition of pancreatic lipase by WSC

As shown in Fig. 11, all five WSC samples inhibited the pancreatic lipase activity dose-dependently in the concentration range of 0–100 µg/mL in the assay system. When the substrate concentration was less than 60 µg/mL, the inhibitory activities of WSC3, WSC4 and WSC5 were better than those of WSC2 and WSC1. There was no significant difference between the activities of WSC3, WSC4 and WSC5. When the substrate concentration was more than 60 µg/mL, the difference became clear, as WSC3 showed the biggest inhibition rate (13.45%). WSC4 was the second largest, with an inhibition rate of 12.19%. WSC5 was the third, with an inhibition rate of 10.50%. WSC2 and WSC1 were the least, with inhibition rates of 7.12% and 4.59%, respectively.

In recent years, pancreatic lipase has become a valid target in the treatment of obesity, which has also attracted great interest (*Wilcox et al., 2014*). However, effective inhibitors with fewer side effects werestill rare. As a dietary fiber, chitosan was reported to be able

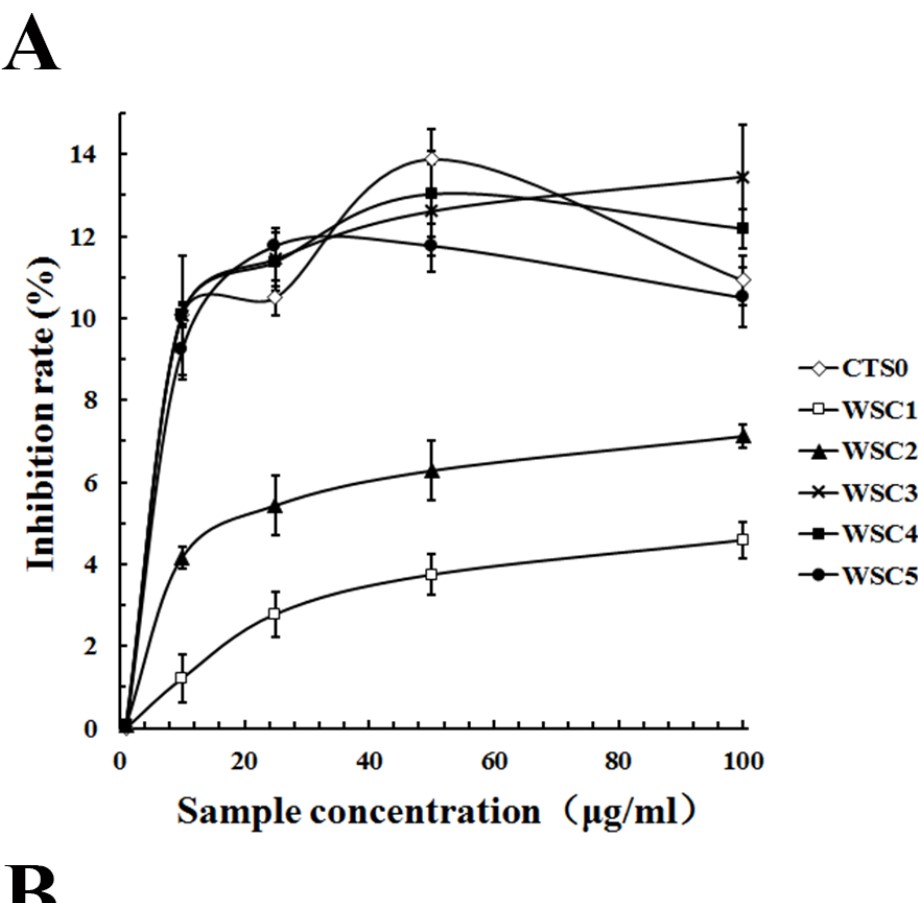

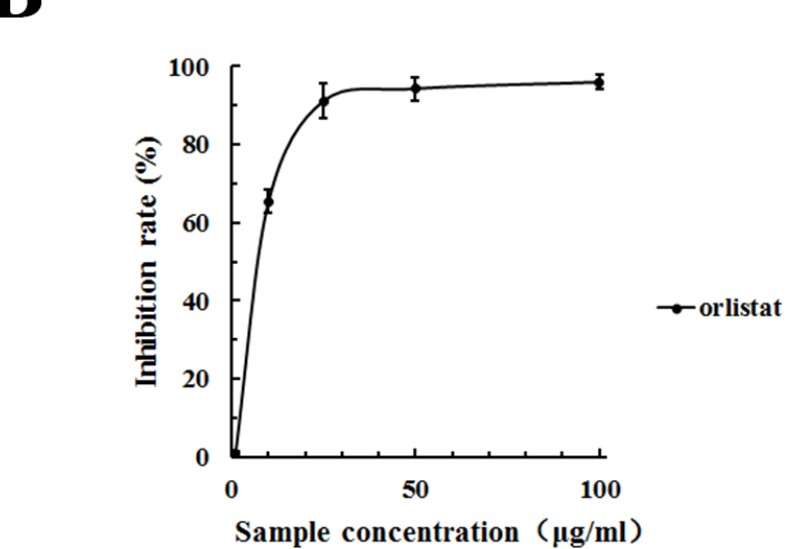

**Figure 11** **Inhibition rate (%) of PL by WSC1-5, CTS0 and orlistat.** The inhibition rate (%) of PL by (A) WSC1-5, CTS and (B) orlistat were determined by colorimetric methods using PNPB as the chromogenic substrate. Data are expressed as the mean ± SD from triplicate experiments. Different small letters next to values indicate significant differences ($p < 0.05$).

to inhibit pancreatic lipase to some extent (*Tsujita et al., 2007*). Great attention and lots of work have been focused on its inhibitory effects. For example, studies of *Sumiyoshi & Kimura (2006)* suggested that the lipid-lowering effect of WSC with low Mw was better than the original chitosan and might be mediated by a decrease in the absorption of dietary lipids (triacylglycerol and cholesterol) from the small intestine as a result of the inhibition of pancreatic lipase activity. Studies by *Han, Kimura & Okuda (1999)* suggested that the site of the pancreatic lipase inhibitory action of WSC mightnot be the enzyme but its substrate. The inhibitory effect of WSC on pancreatic lipase was also suggested to be related to its Mw. For example, WSC with an average molecular mass of 46,000 Da was proven to be a stronger inhibitor than the original chitosan, while WSC with an average molecular mass of 10,000 Da was proven to be a weaker inhibitor than the original chitosan (*Lunagariya et al., 2014*).

Although the highest inhibition rate in this study was just 13.45%, which was far less than the effect of the positive control, orlistat. Samples used in this study were natural marine active substances with fewer side effects, while orlistat was a synthetic chemical that was reported to have some gastrointestinal side effects and safety risks (*Wilcox et al., 2014*). Studies by *Sumiyoshi & Kimura (2006)* proved that WSC did not cause liver damage with the elevation of glutamic oxaloacetic transaminase and glutamic pyruvic transaminase or kidney damage with the elevation of blood nitrogen urea. Therefore, it was suggested that WSC was a safe functional food with great potential for application in anti-obesity efforts. Given that the underlying mechanism of WSC's inhibitory effect on pancreatic lipase is still not proven, the influence of the Mw has not been well studied and few animal or human studies have been performed, further studies are urgently needed.

## CONCLUSION

The five WSC samples used in this study exhibited great fat- and cholesterol-binding capacities. In addition, there was a significant correlation between the binding capacity and Mw of WSC, as WSC2 ($\sim$3,000 Da) showed the highest fat-binding and cholesterol-binding capacities (7.08 g g$^{-1}$ and 63.48 mg g$^{-1}$, respectively). These capacities were much higher than those of cellulose, which were only 0.41 g g$^{-1}$ and 5.95 mg g$^{-1}$, respectively. In addition, the binding abilities declined as the Mw increased or decreased from 3,000 Da. Meanwhile, all WSC samples were proven to be able to inhibit pancreatic lipase activity to some extent. WSC3, WSC4 and WSC5 had higher inhibitory activities than WSC1 and WSC2 at all the tested sample concentrations.

In view of these findings, we speculate that adsorption, electrostatic binding and entrapment of the cholesterol, fat, sterols and triglycerides in food is an important anti-obesity mechanism of WSC. Given that WSC is a natural marine active substance with fewer side effects than orlistat, we believe that it will be a very promising bioactive substance in the field of obesity management. To maximize the development and utilization of WSC, further studies are necessary to investigate its underlying mechanism, anti-obesity efficacy, and bioavailability in animal and human subjects.

### Funding

This work was supported by the NSFC-ShandongJoint Fund (U1606403) and major projects of independent innovation, Shandong (2013CXB80203). The funders had no role in study design, data collection and analysis, decision to publish, or preparation of the manuscript.

### Grant Disclosures

The following grant information was disclosed by the authors:
NSFC-ShandongJoint Fund: U1606403.
Major projects of independent innovation, Shandong: 2013CXB80203.

### Competing Interests

The authors declare there are no competing interests.

### Author Contributions

- Qiu Jin conceived and designed the experiments, performed the experiments, analyzed the data, contributed reagents/materials/analysis tools, wrote the paper, prepared figures and/or tables, reviewed drafts of the paper.
- Huahua Yu conceived and designed the experiments, contributed reagents/materials/-analysis tools, reviewed drafts of the paper.
- Xueqin Wang and Kecheng Li give guides to the experiment.
- Pengcheng Li contributed reagents/materials/analysis tools, reviewed drafts of the paper.

### Data Availability

The raw data has been supplied as Data S1.

### Supplemental Information

Supplemental information for this article can be found online at http://dx.doi.org/10.7717/peerj.3279#supplemental-information.

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
