# Peer review of "Effect of the molecular weight of water-soluble chitosan on its fat-/cholesterol-binding capacities and inhibitory activities to pancreatic lipase"

_PeerJ, doi:10.7717/peerj.3279_

## Round 0.1 · original submission · Major Revisions

Please respond in a point by point manner to the comments from the reviewers, and follow their instructions in the new revision.

Reviewer 1 ·

Basic reporting

The english has to be improved. There are different not clear expresions or ambiguous.

Figures are not all relevant. Figure 2 shows the separation in function of hydrolysis time. The peaks are not different in the different subfigures (between 8 and 12 min). In this way, the caption shows Mw and in the figure only retention times are done. This is the same in other captions where the figure is not showing that.

Experimental design

Experimental procedure has to be better characterized. E.g. separation column is TSKgel® G3000PWXL HPLC Column phase polymethacrylate, hydroxylated, L × I.D. 30 cm × 7.8 mm, 7 μm particle size, and the technical specifications are “Use TSKgel G3000PWxl columns for the analysis of water-soluble linear polymers with molecular weights up to 50,000 Da. The smaller particle size of PWxl-type columns provide 1.7 times higher resolution than their PW counterpart, making PWxl-type columns more suitable for analytical purposes”. How is it possible to use for the separation of polymers among 1294 and 9700 Da?. In this way, the calibration with dextrans is not valid because the retention in the column is via hydrogen bonds and this is different with molecules with hydroxyl groups and hydroxyl and amine groups. To consider different Mws mass spectrometry has to be used.

Other example: Line 181: cooling in ice for 10 minutes do not stop the reaction.

It is difficult to think that the chemical hydrolysis is able to produce pure polymers from a bigger chitosan

Validity of the findings

The mayor impact is the effect of the mw in the "in vitro" activity, the rest is previously reported by other authors.

Additional comments

Considering the problems with the separation and characterization of the obtained polymers, different experiments would be necessary to publish the manuscript. A more detailled experimental part is required and additional experiments of chromatography and mass-spectrometry are required. This is fundamental because is the base of all the reported work.

Reviewer 2 ·

Basic reporting

This is a fine article which study the potential of chitosan as an putative anti obesity drug. The field of the study is relevant and its importance is well supported in a well structured Introduction section that provides sufficient refrences. Some of the results are of relevance and well depicted in the figures.I suggest the authors to avoid ambiguous language (i.e. line 63 "lots of evidence", line 359 "great fat and cholesterol binding capacity") and to revise the English throughout the manuscript.

Experimental design

I have a major concern on the methodological strategy followed to determine the inhibition of the pancreatic lipase by different chitosan molecular weights. In general the results are not presented using enzimological nomenclature and key methodology information is missing to fully understand what the authors have done and how they arrive to their conclusions. For example: what is the source of the lipase? Numerous pancreatic lipase have already been kinetically characterized with natural and synthetic substrates. I was not able to evaluate if the results of inhibition were previously known without the source of the lipase. Also, the authors should define enzymatic units and inhibition rate. The methodological strategy to characterize an inhibitor depends on th type of inhibitor it is. If not known, then the experimental design has to contemplate it. It is extremly important to know at which substrate concentration the enzyme is working in saturating conditions. I assume this was the intention of the authors with the "optimum enzyme concentration experiment". Kinetic parameters (if not previously reported) could have been provided after this experiment. Also for further experiments the authors claim to use this "optimal substrate concentration". For some inhibition experiments it is extremely important to work under saturating conditions (10 Km).

Validity of the findings

I think the findings related to the cholesterol binding to chitosan of different molecular weight is interesting. The inhibitory effect on the pancreatic lipase will also be once improved.

Additional comments

I recommend the authors to streghthen the work related with the inhibition experiments maybe through discussion with an enzymologyst. I also suggest to improve the use of language. I therefore recommend this article for publication after major revision.

---

## Round 0.2 · accepted · Accept

The manuscript has been modified following the reviewers comments, and it can be accepted for publication in the actual form.

Reviewer 1 ·

Basic reporting

The manuscript is acceptable in the actual form. The authors rigorously revised the manuscript. However the mass characterization was not performed, the given reasons were acceptable.

Experimental design

The manuscript is acceptable in the actual form. The authors rigorously revised the manuscript. However the mass characterization was not performed, the given reasons were acceptable.

Validity of the findings

The manuscript is acceptable in the actual form. The authors rigorously revised the manuscript. However the mass characterization was not performed, the given reasons were acceptable.

Additional comments

The manuscript is acceptable in the actual form. The authors rigorously revised the manuscript. However the mass characterization was not performed, the given reasons were acceptable.

Reviewer 2 ·

Basic reporting

The authors have improved the article as suggested by the reviewers and I therefore consider it suitable for publication.

Experimental design

The authors have improved the article as suggested by the reviewers and I therefore consider it suitable for publication.

Validity of the findings

The authors have improved the article as suggested by the reviewers and I therefore consider it suitable for publication.

Additional comments

The authors have improved the article as suggested by the reviewers and I therefore consider it suitable for publication.